# Cost-Efficient Multi-Fidelity Alignment for Large Language Models

## Abstract

Alignment is a critical step in large language model (LLM) post-training. It typically requires human annotations to align the model's output to human preferences, which is prohibitively expensive. This paper proposes a novel approach to reduce the alignment cost. Specifically, we consider multiple levels of alignment with different qualities and response-generating costs, which we refer to as multi-fidelity alignment. We develop a new approach to incorporating the varying levels of response quality to train a language model, aiming to reduce the cost of response collection for alignment while maintaining the performance of the language model. We provide theoretical insights and empirical results to support the effectiveness of the proposed multi-fidelity alignment approach. Lastly, we conduct experiments to corroborate the effectiveness of the proposed approach by comparing its performance with the vanilla alignment methods.

## 1 Introduction

Large language models (LLMs) (Ouyang et al., 2022) have illustrated their power in various tasks, such as text generation (Dathathri et al., 2019), question answering (Su et al., 2019), and image captioning (Devlin et al., 2015). However, to harness its power, one often needs to align these models to human preference so that the generated content meetshuman's expectation. More specially, the alignment either asks the human to demonstrate the task, e.g., write out the answer of a given question for a language model, or to evaluate the outputs of generative models, e.g., rank multiple answers according to human preference. Although crucial in LLMs, alignment is a very expensive task due to the requirement of extensive human participation, and the cost of alignment is a significant challenge in training LLMs (OpenAI, 2023).

Recent progress in alignment shows that weaker supervision or alignment can make a "strong" model (with a large number of model parameters) perform better than the "weak" one that provides the supervision (Burns et al., 2023), a.k.a, "weak-to-strong". Specifically, Burns et al. (2023) corroborates that the output model's performance of using GPT-2's response to align a model with the infrastructure of GPT-4 is similar to GPT-3.5 (exceeding GPT-2). This observation implies that the alignment from a low level of expertise can help train a strong generative model.

With this "weak-to-strong" observation in mind, this paper proposes a novel approach to reducing the alignment costs. Specifically, we consider multiple levels of alignment with different qualities and response-generating costs, which we refer to as multi-fidelity alignment. For example, a high-fidelity alignment can refer to collecting responses or preferences from human experts, while a low-fidelity alignment may correspond to collecting responses or preferences from a language model weaker than human experts. With these alignments of varying quality levels, we propose a new approach to incorporating them to train a language model efficiently. We aim to reduce the cost of response collection for alignment while maintaining the performance of the language model. We note that this cost improvement is from the perspective of response collection, not the computational cost of training the language model, which was the focus of some previous works, like direct preference optimization (DPO) (Rafailov et al., 2024).

This paper is organized as follows,

- In Section 2, we propose a mathematical formulation of the win rate (WINRATE) used in the preference-based training, e.g., alignment training of LLMs and provide a theoretical analysis of the win rate and its empirical estimation.

- In Section 3, we devise a novel multi-fidelity alignment algorithm to reduce the cost of response collection for alignment of large language models.

- In Section 4, we provide a theoretical analysis for the cost complexity upper bound of the multi-fidelity alignment algorithm.

- In Section 5, we conduct experiments to corroborate the effectiveness of the proposed approach by comparing its performance with the vanilla alignment methods.

## 1.1 RELATED WORK

**Alignment**    To align the powerful generative models to human preference, various alignment methods have been proposed (Ouyang et al., 2022; Rafailov et al., 2024; Chen et al., 2024; Hong et al., 2024). Reinforcement learning with human feedback (RLHF) (Ouyang et al., 2022) is one of the most popular method to align a language model. This method consists of three steps: (1) use collected response to fine-tune the language model, (2) use the supervised fine-tuning (SFT) model to generate responses, ask human experts to label the preference of these responses, and then uses the preference labels to train a reward model, and (3) use the estimated reward model to align the LLM via proximal policy optimization (PPO). In the second step, one alternative is to ask another advanced large language model to label the preference, which is known as reinforcement learning with artificial intelligence feedback (RLAIF) (Bai et al., 2022; Lee et al., 2023). However, both the RLHF and RLAIF rely on an intermediate reward model, and the PPO training is computational expensive. Motivated to remove the intermediate reward model, Rafailov et al. (2024) proposes a direct policy optimization (DPO) method to align the language model to human performance without the need of an intermediate reward model. DPO is a more efficient method to align the language model and can be optimized via simple supervised learning methods instead of the PPO in reinforcement learning. Following the spirit of DPO, many very recent new alignment methods have been proposed, such as self-play fine-tuning (SPIN) (Chen et al., 2024) and monolithic odds ratio preference optimization (ORPO) (Hong et al., 2024). While these reward-model free methods are more efficient in the perspective of computational cost, they still require a large number of human responses to align the language model, which is also expensive. This paper aims to further reduce the cost of the response collection in alignment.

**Multi-Fidelity**    Multi-fidelity optimization is a general optimization framework that aims to optimize an expensive objective function by using multiple levels of fidelity approximations (Bonfiglio et al., 2018; Zheng et al., 2013; Forrester et al., 2007; Huang et al., 2006). Later on, the multi-fidelity approach is extended to the literature of AutoML, e.g., HyperBand (Li et al., 2017) and BOHB (Falkner et al., 2018). Meanwhile, the multi-fidelity approach is also studied in the sequential decision making literature, e.g., in multi-armed bandits (Kandasamy et al., 2016) and Bayesian optimization (Kandasamy et al., 2017; 2019). Very recently, the multi-fidelity best arm identification (MF-BAI) problem is studied by Poiani et al. (2022; 2024); Wang et al. (2023). This paper is the first to study the multi-fidelity approach in the context of alignment of large language models.

## 2 MODEL FORMULATION

### 2.1 BASIC MODEL

Denote $\boldsymbol{\theta} \in \mathbb{R}^N$ as the parameters in a large language model (LLM) $f : \mathcal{X} \rightarrow \mathcal{Y}$, where $N$ is often a dramatically large integer, e.g., $N \geqslant 10^{11}$ in GPT-4 (OpenAI, 2023), $\mathcal{X}$ is the set of input tokens (i.e., prompts/questions), and $\mathcal{Y}$ is the set of output tokens (i.e., answers). Specifically, denote $f(x; \boldsymbol{\theta})$ as the output of the given generative model with parameters $\boldsymbol{\theta}$ and input token $x$, and we use $f(\cdot; \boldsymbol{\theta})$ to represent the LLM model $f$ with parameters $\boldsymbol{\theta}$. Given $M \in \mathbb{N}^+$ fidelities that one can query responses for alignment, we denote $\{g^{(1)}, g^{(2)}, \ldots, g^{(M)}\}$ as the LLMs with different levels of expertise (qualities), where $g^{(m)}$ is able to generate output with quality in level $m$. We denote the cost of querying the fidelity $m$ model once as $\lambda_{\text{cost}}^{(m)}$.

## 2.2 WINRATE METRIC

The alignment task is often conducted via training from input preference pairs. The preference pair consists of winning and losing responses, depending on the response qualities. To formally describe the response qualities, we introduce a metric WINRATE to evaluate the performance of two LLMs. It represents the performance distance between two LLMs $f_1$ and $f_2$ under preference comparison function $F$, formally defined as follows,

$$\text{WINRATE}(f_1, f_2, F) :=$$
$$\mathbb{E}_{x \sim \mathcal{X}, f_1, f_2} \mathbb{E}_F [\mathbb{1}\{F(f_1(x), f_2(x)) = \text{Win}\} + 0.5 \cdot \mathbb{1}\{F(f_1(x), f_2(x)) = \text{Tie}\}], \quad (1)$$

where $F$ is the comparison function that outputs "Win" (resp., "Tie" / "Lose") if the response from the first model is better than (respectively, close to / worse than) that of the second one,[1] $\mathbb{1}\{\cdot\}$ denotes an indicator function, the outer expectation is taken over the randomness of the input token $x$ and the LLMs $f_1, f_2$, and the inner expectation is taken over the randomness of the comparison function $F$. Without loss of generality, we label the $M$ fidelity models such that for any two fidelities $m > m'$, $\text{WINRATE}(g^{(m)}, g^{(m')}) > 50\%$, denoted as $g^{(m)} \succ g^{(m')}$.

To empirically estimate the WINRATE metric, one can collect a set $\mathcal{D}(f_1, f_2, F, n)$ of $n \in \mathbb{N}^+$ response pairs of the two LLMs. Denote the tuple $(y_1, y_2, x, F)$ as one element of the dataset $\mathcal{D}$, where $y_1$ and $y_2$ are the outputs of $f_1$ and $f_2$ given the input prompt $x$, respectively, and $F$ is the comparison function used to compare the two responses $y_1, y_2$. Formally, WINRATE can be estimated by calculating the *win frequency* as follows,

$$\text{WINFREQ}(\mathcal{D}) := \frac{\sum_{(y_1, y_2, x, F) \in \mathcal{D}} (\mathbb{1}\{F = \text{Win}\} + 0.5 \cdot \mathbb{1}\{F = \text{Tie}\})}{|\mathcal{D}|},$$

and $\text{WINRATE}(f_1, f_2) = \lim_{n \to \infty} \text{WINFREQ}(\mathcal{D}(f_1, f_2, F, n))$.

Given the definitions of WINRATE and WINFREQ, the following lemma shows how accurate the empirical WINRATE (frequency) can estimate the idealistic WINRATE. All formal proofs of the paper are deferred to the appendix.

**Lemma 1** (From WINFREQ to WINRATE). *Assume $\mathcal{D}(f_1, f_2, F, n)$ is a set of $n$ samples generated by comparing LLMs $f_1$ and $f_2$ with fidelity $M$ via uniform randomly selected input tokens from $\mathcal{X}$. Then, with probability at least $1 - \delta$ for a confidence parameter $\delta \in (0, 1)$, we have*

$$|\text{WINFREQ}(\mathcal{D}(f_1, f_2, F, n)) - \text{WINRATE}(f_1, f_2, F)| \leqslant \sqrt{\frac{1}{n} \log \frac{2}{\delta}}.$$

Next, Lemma 2 presents the smoothness of the WINRATE metric with respect to the parameters of the LLMs. It shows that the parameter difference between two LLMs is upper bounded by the difference in the WINRATE metric of the two LLMs.

**Lemma 2** (Smoothness of WINRATE). *Utilize gradient ascent to maximize the function $\text{WINRATE}(f(\cdot, \boldsymbol{\theta}), g^{(m)})$ beginning with an initial parameter $\boldsymbol{\theta}_0$, and conclude at $\boldsymbol{\theta}_1$. Assuming $\text{WINRATE}(f(\cdot, \boldsymbol{\theta}), g^{(m)})$ is locally concave with respect to a local maximizer $\boldsymbol{\theta}_2$, we have, for some constant parameter $\alpha > \eta/\epsilon$,*

$$\|\boldsymbol{\theta}_0 - \boldsymbol{\theta}_1\| \leqslant \alpha |\text{WINRATE}(f(\cdot; \boldsymbol{\theta}_0), g^{(m)}) - \text{WINRATE}(f(\cdot; \boldsymbol{\theta}_1), g^{(m)})|.$$

Lastly, we introduce the following assumption to illustrate that the relationship of WINRATE when comparing every two of three LLMs.

**Assumption 3.** *For any two fidelities $m > m'$ and any parameter $\boldsymbol{\theta}$, there exists a constant $\beta > 0$ such that $|\text{WINRATE}(f(\cdot, \boldsymbol{\theta}), g^{(m)}) - \text{WINRATE}(f(\cdot, \boldsymbol{\theta}), g^{(m')})| \leqslant \beta \text{WINRATE}(g^{(m)}, g^{(m')})$.*

Assumption 3 is an analogy of the triangle inequality (though with a scaling factor $\beta$) for the WINRATE metric. Because there should exist a evaluation function $W : f \mapsto \text{score} \in \mathbb{R}$ for any LLM, the WINRATE of one LLM $f_1$ against another $f_2$ can be calculated as the difference between the evaluation scores of the two LLMs, i.e., $f_1(W) - f_2(W)$. This underline evaluation function $W$ implies that the WINRATE metric is separable and thus fulfills the triangle inequality.

---

[1] In practice, one can either ask human experts or another advanced LLM model to compare the outputs of two models. The experiments of this paper use GPT-4 as the comparison function $F$.

## 2.3 MULTI-FIDELITY ALIGNMENT PROBLEM AND OBJECTIVE

In this paper, we study a multi-fidelity alignment problem, formally described in Procedure 1. It proceeds in rounds, where in each round, it selects a fidelity $m_t$ and a batch size $n_t$ to query responses from the fidelity $m_t$ model as well as the trained LLM $f(\boldsymbol{\theta}_t)$. Then, based on these responses, it constructs a dataset $\mathcal{D}_t$ and aligns the LLM $f(\boldsymbol{\theta}_t)$ by updating its parameter $\boldsymbol{\theta}_t$.

---

**Procedure 1** Multi-Fidelity Alignment

---

1: **for** each decision round $t = 1, 2, \ldots$ **do**
2:     Pick a fidelity $m_t$ and a batch size $n_t$ to query responses
3:     Query $n_t$ responses from the fidelity $m_t$ model as well as the trained LLM $f(\cdot; \boldsymbol{\theta}_t)$
4:     Construct the dataset $\mathcal{D}_t$ from the responses
5:     Align the LLM $f(\boldsymbol{\theta}_t)$ via updating its parameter $\boldsymbol{\theta}_t$ based on the dataset $\mathcal{D}_t$
6: **end for**

---

The objective is to devise an algorithm $\mathcal{A}$ such that it can efficiently align the LLM $f(\cdot; \boldsymbol{\theta})$ to achieve a high WINRATE with the highest fidelity model $g^{(M)}$ while minimizing the total cost of querying responses from the fidelity models and the LLM. Formally, the objective is to cost-efficiently find parameters $\boldsymbol{\theta}$ such that WINRATE$(f(\cdot; \boldsymbol{\theta}), g^{(M)}) \geqslant \gamma$ for some given parameter $\gamma > 0$ with a probability of at least $1 - \delta$ while minimizing the total cost $\mathbb{E}\left[\sum_t n_t \lambda_{\text{cost}}^{(m_t)}\right] =: \Lambda$.

## 3 ALGORITHM

This section presents a multi-fidelity alignment algorithm (Algorithm 2). Given a set of $M$ fidelities to query responses with different costs, the algorithm aims to train a large language model (LLM) with small costs.

We consider a step-by-step alignment process, where the algorithm starts with the lowest fidelity and gradually increases the fidelity until the stopping criterion is met. The stopping criterion consists of two parts (inverse of Line 2): either the WINRATE between the trained LLM $f(\cdot; \boldsymbol{\theta}_t)$ and the fidelity expert $g^{(m)}$ exceeds a threshold $\gamma$ or the LLM parameters $\boldsymbol{\theta}_t$ converge. Especially, we use WINFREQ$(f, g)$ to empirical estimate the WINRATE between two models/experts $f$ and $g$ (Line 2). After the algorithm finishes the training at the last fidelity $M$, it outputs the trained LLM and its parameters $f(\cdot; \boldsymbol{\theta}_t)$.

At each fidelity $m$, the algorithm maintains a dataset $\mathcal{D}^{(m)}$, updated in a batch manner. For each batch, the algorithm, by randomly picking $B \in \mathbb{N}^+$ prompts from a given set $\mathcal{X}$, collects $B$ response pairs from both the trained LLM $f(\cdot; \boldsymbol{\theta}_t)$ and fidelity expert $g^{(m)}$ (Lines 5–8), where the subroutine Generate$(x; f)$ means to generate one response for prompt $x$ from a LLM or expert $f$. Then, the algorithm concatenates these new responses pairs to the dataset $\mathcal{D}^{(m)}$ (Line 9). Lastly, with this extended dataset $\mathcal{D}^{(m)}$, the algorithm updates the LLM parameters $\boldsymbol{\theta}_t$ using directed preference optimization (DPO) (Rafailov et al., 2024) (Line 11).

## 4 THEORETICAL ANALYSIS

In this section, we provide a theoretical analysis of the multi-fidelity alignment algorithm (Algorithm 2). In order to analysis the total cost of the algorithm, we first make an assumption on the convergence rate of the DPO training process (used in Line 11). Then, we provide a lemma to upper bound the total cost of the algorithm. Finally, we provide a theorem to upper bound the cost complexity of the algorithm.

**Assumption 4** (Linear convergence rate). $n = \Theta\left(\log \frac{\|\boldsymbol{\theta}_0 - \boldsymbol{\theta}(n)\|}{1 - \gamma}\right)$ *iterations are sufficient to achieve* WINRATE$(f(\cdot, \boldsymbol{\theta}(n)), g^{(m)}) \geqslant \gamma$ *(for any* $\gamma \leqslant \lim_{n \to \infty}$ WINRATE$(f(\cdot, \boldsymbol{\theta}(n)), g^{(m)}))$, *where* $\boldsymbol{\theta}_0$ *is the initial parameter at the beginning of training at fidelity* $m$, *and* $\boldsymbol{\theta}(n)$ *is the parameter after* $n$ *iterations.*

---

**Algorithm 2** Multi-Fidelity Alignment

---

**Input:** confidence parameter $\delta$, error parameter $\epsilon$, WINRATE stopping threshold $\gamma \in (0,1)$, and prompt set $\mathcal{X}$, training base LLM $f(\cdot; \boldsymbol{\theta})$, multi-fidelity LLM/human experts $g^{(m)}$ for $m = 1, 2, \ldots, M$, and batch size $B \in \mathbb{N}^+$

**Initialize:** fidelity index $m \leftarrow 1$, time $t \leftarrow 1$, LLM parameter $\boldsymbol{\theta}_1$, and dataset $\mathcal{D}^{(m')} \leftarrow \emptyset$ for all fidelities $m' = 1, \ldots, M$

1: **for** each fidelity $m = 1, 2, \ldots, M$ **do**
2:   **while** WINFREQ$(f(\cdot; \boldsymbol{\theta}_t), g^{(m)}) < \gamma$ and $\|\boldsymbol{\theta}_t - \boldsymbol{\theta}_{t-1}\| > \epsilon$ **do**
3:     $t \leftarrow t + 1$                                                    ▷ Update parameter index
4:     **for** $i = 1, \ldots, B$ **do**                              ▷ Collect new responses
5:       Randomly pick a prompt $x_i$ from dataset $\mathcal{X}$
6:       $y_{1,i} \leftarrow$ Generate$(x_i; f(\cdot; \boldsymbol{\theta}_t))$       ▷ Response from trained LLM
7:       $y_{2,i} \leftarrow$ Generate$(x_i; g^{(m)})$                    ▷ Response from expert
8:       $F_i \leftarrow$ Compare$(y_{1,i}, y_{2,i}; x_i, F)$
9:       $\mathcal{D}^{(m)} \leftarrow \mathcal{D}^{(m)} \cup \{y_{1,i}, y_{2,i}, x_i, F_i\}$
10:     **end for**
11:     $\boldsymbol{\theta}_t \leftarrow$ DPO_Update$(f, \boldsymbol{\theta}_{t-1}, \mathcal{D}^{(m)})$                    ▷ DPO Training
12:   **end while**
13: **end for**

**Output:** trained LLM and its parameters $f(\cdot; \boldsymbol{\theta}_t)$

---

The linear convergence rate in Assumption 4 is a common assumption in the optimization literature, and it is also validated validated in deep learning theory literature (Allen-Zhu et al., 2019) for deep neural network (DNN) and convolutional neural network (CNN) training.

From the pseudo-code of Algorithm 2, we can see that the cost of the algorithm is mainly from the response collection at each fidelity. The cost of generating a response from the fidelity expert $g^{(m)}$ is denoted as $\lambda_{\text{cost}}^{(m)}$. The size of the dataset constructed at fidelity $m$ is denoted as $|\mathcal{D}^{(m)}|$. The total cost of the algorithm is the sum of the costs at each fidelity, that is, $\sum_{m=1}^{M} \lambda_{\text{cost}}^{(m)} |\mathcal{D}^{(m)}|$. By upper bounding the size of the dataset $|\mathcal{D}^{(m)}|$ at each fidelity, we propose an upper bound for the total cost of the algorithm in Theorem 5.

**Theorem 5.** *Assuming the triangle inequality property of* WINRATE *(Assumption 3), and the linear convergence rate (Assumption 4), the cost complexity of Algorithm 2 is upper bounded as follows, with a probability of at least $1 - \delta \sum_{m \in \mathcal{M}} |\mathcal{D}^{(m)}|$,*

$$\Lambda_{\text{MF}} \leqslant O\left( \sum_{m=1}^{M} \lambda_{\text{cost}}^{(m)} \left( \log \frac{2\alpha \cdot \text{rad} + \alpha\beta \, \text{WINRATE}(g^{(m)}, g^{(m-1)})}{1 - \gamma - \text{rad}} + B \right) \right), \qquad (2)$$

*where $g^{(0)} := f(\cdot; \boldsymbol{\theta}_0)$ is the LLM $f$ with the initial parameter $\boldsymbol{\theta}_0$, the parameters $\delta \in (0,1)$ and* rad $> 0$ *depend on the* WINFREQ *calculation in Line 2 of Algorithm 2, and parameters $\alpha, \beta, \gamma$ are defined in Lemma 2 and Assumptions 3 and 4, respectively.*

The confidence level $\delta$ and the confidence radius rad depends on the number of evaluation samples used in the WINFREQ calculation in Line 2 of Algorithm 2, representing the variance in estimating the actual WINRATE. The larger the number of samples, the smaller the $\delta$ and the rad as quantified in Lemma 1. The $\delta$ and the rad are used to ensure the probability of the WINFREQ calculation in Line 2 of Algorithm 2 is at least $1 - \delta \sum_{m \in \mathcal{M}} |\mathcal{D}^{(m)}|$. The additive $MB$ terms in (2) is the cost of generating a response from the LLM $f(\cdot; \boldsymbol{\theta}_t)$ at each fidelity, where $B$ is the batch size used in the response collection at each fidelity.

One vanilla alignment approach is the high-fidelity (HF) alignment, where one only uses the responses collected from the highest fidelity expert $g^{(M)}$ to train the LLM $f$. With the same stopping condition as in Algorithm 2, the cost complexity of the high-fidelity alignment algorithm is upper bounded as follows,

$$\Lambda_{\text{HF}} = O\left( \lambda_{\text{cost}}^{(M)} \left( \log \frac{2\alpha \cdot \text{rad} + \alpha\beta \, \text{WINRATE}(g^{(M)}, g^{(0)})}{1 - \gamma - \text{rad}} + B \right) \right). \qquad (3)$$

Table 1: WINRATE of our LLM and fidelities

| Baseline Model | Candidate Model | Win rate |
|:---:|:---:|:---:|
| Vicuna | Llama | 0.785 |
| Llama | Mistral | 0.519875 |
| Mistral | Qwen | 0.695 |

The most favorable scenario for the multi-fidelity alignment algorithm is when the cost of the highest fidelity expert is much larger than the cost of the lower fidelity experts, i.e., $\lambda_{\text{cost}}^{(M)} \gg \lambda_{\text{cost}}^{(m)}$ for all $m < M$. For example, one extreme case is that the fidelity experts $g^{(m)}$ with $m < M$ other than the highest fidelity are all free to collect responses from (i.e., $\lambda_{\text{cost}}^{(m)} = 0$ for $m < M$), e.g., they are all language models that are already available. Then, the cost complexity of the multi-fidelity alignment algorithm is reduced to $\Lambda_{\text{MF}} = O\left(\lambda_{\text{cost}}^{(M)} \left(\log \frac{2\alpha \cdot \text{rad} + \alpha\beta \, \text{WINRATE}(g^{(M)}, g^{(M-1)})}{1 - \gamma - \text{rad}} + B\right)\right)$. As long as the $\text{WINRATE}(g^{(M)}, g^{(M-1)}) < \text{WINRATE}(g^{(M)}, g^{(0)})$, the multi-fidelity alignment algorithm would be more cost-effective than the high-fidelity alignment algorithm. In practice, the cost of the highest fidelity expert is usually much larger than the cost of the lower fidelity experts, which is close to the extreme example above, and hence, the multi-fidelity alignment algorithm is more cost-effective than the high-fidelity alignment algorithm.

## 5 EXPERIMENTS

This section empirically evaluates the effectiveness of multi-fidelity alignment to obtain a better LLM under a given budget. First, we employ 3 fidelities with increased performance to train our LLM step by step, and compare it with directly using DPO to train our LLM. We choose the best model derived from two methods and evaluate the model's performance on the test dataset. Our results show that if we allocate part of the budget to lower fidelity responses, we can derive an LLM with better performance, i.e., better use of the budget. Before presenting these results, we describe the experiment setup.

### 5.1 SETUP

**Datasets.** In this study, we adopt **rm-static**[2], a split of Anthropic's Helpful Harmless dataset, as our dataset. In our experiment, we shuffle the dataset and reserve 200 samples for validation and 500 samples for testing.

**Hyper-parameter setting.** During policy training, we set maximum length to 256, adopt top-p sampling with $p = 0.9$ and temperature set to 0.1. When testing response generation, we set maximum length to 256, $p = 0.95$ and restrict to top-k=50 in top-p sampling, and temperature to 0.1. For DPO training, we set its learning rate to 5e-6, using cosine learning rate schedule and leave 10 percent of steps for warm up. For the cost setting of different fidelities, we set it to 1:4:16 for each response they generate. For upgrade criterion, we use the win rate metric: if our LLM's win rate against fidelity exceeds 0.3, we proceed to the next fidelity. For the final fidelity, training is halted when the win rate declines compared to the previous iteration. This criterion is chosen for two reasons. First, it minimizes unnecessary costs associated with not utilizing the newly trained LLM. Second, since the LLM's response is automatically designated as the rejected response in DPO training, a lower win rate in the current LLM supports better performance in the subsequent training process. For batch size, we conducted training using batch sizes B of 512 and 1024, with each experiment repeated twice.

**Evaluation metrics.** We employ our win rate metric to assess model performance. First, we collect the responses of two models to a variety of prompts, designating one model as the baseline, typically a fidelity, and the other as the candidate model, usually derived from training. We then utilize GPT-4o to evaluate the responses in a pairwise manner, calculating the total number of instances where the candidate model outperforms the baseline. The win rate of the candidate model, denoted as

---

[2]https://huggingface.co/datasets/Dahoas/rm-static

Table 2: WINRATE (calculated in Line 2 of Algorithm 2) and fidelity shift trajectory in multi-fidelity alignment with rm-static dataset with batch size $B = 1024$ samples per iteration in two traces

| Fidelity | Alignment Trace One | | Alignment Trace Two | |
| | Candidate model | WINRATE | Candidate model | WINRATE |
| --- | --- | --- | --- | --- |
| $g^{(1)}$: Llama | Vicuna | 0.215 | Vicuna | 0.238 |
| | step1,iter1 | 0.225 | step1,iter1 | 0.241 |
| | step1,iter2 | **0.354** | step1,iter2 | 0.289 |
| | - | - | step1,iter3 | **0.309** |
| $g^{(2)}$: Mistral | step1,iter2 | **0.318** | step1,iter3 | 0.284 |
| | - | - | step2,iter1 | **0.386** |
| | vanilla ($m = 2$) | 0.269 | vanilla ($m = 2$) | 0.256 |
| $g^{(3)}$: Qwen | step1,iter2 | 0.163 | step2,iter1 | **0.229** |
| | step3,iter1 | 0.204 | step3,iter1 | 0.210 |
| | step3,iter2 | **0.233** | - | - |
| | step3,iter3 | 0.213 | - | - |
| | vanilla ($m = 3$) | 0.191 | vanilla ($m = 3$) | 0.166 |

WINRATE, is determined by dividing the number of wins by the total number of response pairs. A higher win rate indicates superior performance. During training, WINRATE is calculated using a validation dataset, and after identifying the optimal models through our algorithm and DPO, the test dataset is used to compute the final WINRATE.

**Fidelity setting.** We selected **Llama3-8B-Instruct**, **Mistral-7B-Instruct-v0.2**, and **Qwen2-7B-Instruct** as fidelities due to the noticeable performance gap observed on the dataset (as shown in Table 1). To simpilify our descritpions, we refer to these models as **Llama**, **Mistral**, and **Qwen**, respectively. Additionally, we selected **Vicuna-7b-v1.5** as our LLM for training, denoted as **Vicuna**. We define step **m**,iter **n** as the LLM derived from the **nth** iteration using the **mth** fidelity. Specifically, we denote **vanilla**($m = 2$) as the best LLM derived using Mistral under the same budget as in step1 and step2, and **vanilla**($m = 3$) as the best LLM derived using Qwen under the same budget as in the whole step training.

## 5.2 MAIN RESULTS

We present the training trajectories in Table 2 and 3, as well as the testing performance in Table 4. We observed that the experiment results accord with our expectations. In step 1 and step 2, we switch to the next fidelity when our LLM's WINRATE exceeds 0.3 and note our models outperform vanilla ($m = 2$) in both validation and test datasets. In step 3, training is halted when the LLM's WINRATE begins to decline, and the LLM with the highest WINRATE against Qwen is selected as our best LLM. Our results indicate that our LLMs outperform vanilla ($m = 3$) in both validation and test datasets with a batch size of $B = 1024$, and our LLMs surpass vanilla ($m = 3$) in the validation dataset with a batch size of $B = 512$. We will analyze why our LLM is slightly inferior to vanilla($m = 3$) in test dataset next.

**Discussion.** In Table 4, we observe that the performance is constrained when the sample size is limited to 512 per iteration. Based on this, we conclude the following suggestions for applying our method. First, GPT evaluation and prompts have variances. When the increment of performance is small, GPT may mislead us to move to the next fidelity or stop training. Second, it's difficult to align the LLM's distribution to fidelity's distribution when using small amount of samples. The distribution may end up in an inferior position which leads to the drop of performance. Combining the first two points, we give two suggestions to get a satisfactory result. First, we can train on a larger batch size to get greater improvement and give our LLM enough time to align its distribution with fidelities. Second, we can enlarge our validation dataset for more accurate results. Third, automatically labeling our LLM's response as rejected response may not be effective when LLM's WINRATE arise. As the performance of the student model improves, it may generate responses that surpass the current fidelity, potentially leading to incorrect preference labels for the DPO training. The more incorrect labeling, the more likely the performance will decrease. To mitigate this limitation, we test different quality of rejected responses for DPO training and find that rejected responses with

Table 3: WINRATE (calculated in Line 2 of Algorithm 2) and fidelity shift trajectory in multi-fidelity alignment with rm-static dataset with batch size $B = 512$ samples per iteration in two traces

| Fidelity | Alignment Trace One | | Alignment Trace Two | |
|---|---|---|---|---|
| | Candidate model | WINRATE | Candidate model | WINRATE |
| $g^{(1)}$: Llama | Vicuna | 0.215 | Vicuna | 0.250 |
| | step1,iter1 | 0.230 | step1,iter1 | 0.211 |
| | step1,iter2 | 0.299 | step1,iter 2 | 0.226 |
| | step1,iter3 | **0.31** | step1,iter 3 | **0.303** |
| $g^{(2)}$: Mistral | step1,iter3 | 0.275 | step1,iter3 | 0.269 |
| | step2,iter1 | **0.365** | step2,iter1 | **0.349** |
| | vanilla ($m = 2$) | 0.308 | vanilla ($m = 2$) | 0.239 |
| $g^{(3)}$: Qwen | step2,iter1 | **0.186** | step2,iter1 | 0.201 |
| | step3,iter1 | 0.175 | step3,iter1 | **0.209** |
| | - | - | step3,iter2 | 0.159 |
| | vanilla ($m = 3$) | 0.166 | vanilla ($m = 3$) | 0.205 |

Table 4: WINRATE with vanilla as baseline model in test dataset

| Sample size | vanilla($m = 2$) | | vanilla($m = 3$) | |
|---|---|---|---|---|
| | Candidate model | WINRATE | Candidate model | WINRATE |
| 1024 | step1,iter2 | 0.577 | step3,iter2 | 0.513 |
| 1024-2 | step2,iter1 | 0.634 | step2,iter1 | 0.551 |
| 512 | step2,iter1 | 0.546 | step2,iter1 | 0.490 |
| 512-2 | step2,iter1 | 0.590 | step3,iter1 | 0.494 |

relatively poor quality can lead to better performance. Therefore, when we widen the gap between the response pairs, we can obtain greater improvement.

## 6 CONCLUSION

This paper proposed a multi-fidelity alignment algorithm that incorporates responses with varying levels of qualities (fidelities). Theoretically, by formulating a new WINRATE metric for the preference alignment task, the paper rigorously studied the required total cost of the new alignment algorithm, providing insights for practice. Empirically, this paper reports experiments with three fidelities (represented by three LLMs) and validates the superiority of the multi-fidelity algorithm over the standard alignment method in the literature.

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

# A    DEFERRED PROOFS

*Proof of Lemma 1.* The proof is based on the Hoeffding's inequality. Let $X_i =$ $\mathbb{1}\{F(f_1(x_i), f_2(x_i)) = \texttt{Win or Tie}\}$ for $i \in \{1, 2, \ldots, n\}$ where $x_i$ is the $i$-th input token, where $X_i$ can be regarded as a Bernoulli random variable with mean $\text{WINRATE}(f_1, f_2, F)$. Then, $\text{WINFREQ}(\mathcal{D}(f_1, f_2, F, n)) = \frac{1}{n}\sum_{i=1}^n X_i$. By Hoeffding's inequality, we have

$$\mathbb{P}\left(|\text{WINFREQ}(\mathcal{D}(f_1, f_2, F, n)) - \text{WINRATE}(f_1, f_2, F)| \geqslant \epsilon\right) \leqslant 2\exp\left(-n\epsilon^2\right).$$

The proof is complete.  □

*New Proof of Lemma 1 for Separable Assumption.* The proof is based on the Hoeffding's inequality. Notice

$$\text{WINFREQ}(\mathcal{D}(f_1, f_2, F, n)) = \frac{1}{n}\sum_{i=1}^n (W_F(f_1(x_i)) - W_F(f_2(x_i)))$$

$$\text{WINRATE}(f_1, f_2, F) = \mathbb{E}_{x \sim \mathcal{X}}\mathbb{E}_{Y_1 \sim f_1(x), Y_2 \sim f_2(x)}(W_F(Y_1) - W_F(Y_2)),$$

where $x_i$ is the $i$-th input token, and $W_F(Y_1) - W_F(Y_2)$ can be regarded as a $[-1, 1]$-bounded random variable with mean $\mathbb{E}_{x \sim \mathcal{X}}\mathbb{E}_{Y_1 \sim f_1(x), Y_2 \sim f_2(x)}W_F(Y_1) - W_F(Y_2)$. By Hoeffding's inequality, we have

$$\mathbb{P}\left(|\text{WINFREQ}(\mathcal{D}(f_1, f_2, F, n)) - \text{WINRATE}(f_1, f_2, F)| \geqslant \epsilon\right) \leqslant 2\exp\left(-n\epsilon^2/4\right).$$

The proof is complete.  □

*Proof of Lemma 2.* Denote the parameter sequence of the gradient ascent as $\boldsymbol{\theta}_0 = \boldsymbol{\theta}_0^{(1)} \to \boldsymbol{\theta}_1^{(1)} = \boldsymbol{\theta}_0^{(2)} \to \boldsymbol{\theta}_1^{(2)} = \boldsymbol{\theta}_0^{(3)} \to \cdots \to \boldsymbol{\theta}_1^{(k')} = \boldsymbol{\theta}_1 = \boldsymbol{\theta}_0^{(k'+1)} \to \ldots$, where $\boldsymbol{\theta}_0^{(k)}$ is the initial parameter of the $k$-th iteration, and we denote $\boldsymbol{\theta}_1^{(k')} = \boldsymbol{\theta}_1$ as the parameter after $(k')^{\text{th}}$ iteration.

Notice that for each iteration $k$ before stopping, we have

$$\frac{\|\boldsymbol{\theta}_0^{(k)} - \boldsymbol{\theta}_1^{(k)}\|}{|\text{WINRATE}(f(\cdot; \boldsymbol{\theta}_0^{(k)}), g^{(m)}) - \text{WINRATE}(f(\cdot; \boldsymbol{\theta}_1^{(k)}), g^{(m)})|} \leqslant \frac{\eta}{\epsilon},$$

where the last inequality is due to the stop condition $\text{WINFREQ}(\mathcal{D}_{t-1}) - \text{WINFREQ}(\mathcal{D}_{t-2}) \leqslant \epsilon$ in Line 2 of Algorithm 2 is not triggered, and the definition of the learning rate $\eta$.

Therefore, picking any $\alpha^{(m)} > \frac{\eta}{\epsilon}$, for any iteration of the gradient ascent, we have

$$\|\boldsymbol{\theta}_0^{(k)} - \boldsymbol{\theta}_1^{(k)}\| \leqslant \alpha^{(m)}|\text{WINRATE}(f(\cdot; \boldsymbol{\theta}_0^{(k)}), g^{(m)}) - \text{WINRATE}(f(\cdot; \boldsymbol{\theta}_1^{(k)}), g^{(m)})|.$$

Summing up the above inequality for iterations from $k = 1$ to $k'$, we have

$$\|\boldsymbol{\theta}_0 - \boldsymbol{\theta}_1\| \leqslant \sum_{k=1}^{k'} \|\boldsymbol{\theta}_0^{(k)} - \boldsymbol{\theta}_1^{(k)}\| \leqslant \alpha^{(m)}|\text{WINRATE}(f(\cdot; \boldsymbol{\theta}_0), g^{(m)}) - \text{WINRATE}(f(\cdot; \boldsymbol{\theta}_1), g^{(m)})|.$$

□

*Proof of Theorem 5.* The proof is straightforward by applying Lemma **??** and the above analysis. With the double increasing rate, we know that $\sum_{p=1}^{p^{(m)}} B(p) \leqslant 2n^{(m)}$ where $n^{(m)}$ is the number of queries at fidelity $m$ that is enough to achieve $\text{WINFREQ}(\mathcal{D}(f(\boldsymbol{\theta}_t), g^{(m)})) \geqslant \gamma$. Therefore, in the following proof, we assume $\text{WINFREQ}(\mathcal{D}(f(\hat{\boldsymbol{\theta}}^{(m)}), g^{(m)})) = \gamma$, where $\hat{\boldsymbol{\theta}}^{(m)}$ is the final parameters of training at fidelity $m$. Next, we bound the number of queries $n^{(m)}$ by the convergence rate function $r$ and the distance between the two fidelities.

Denote $\hat{\boldsymbol{\theta}}^{(m-1)}$ as the final fidelity at fidelity $m-1$ that fulfills the while-loop's stopping condition $\text{WINFREQ}(f(\cdot; \hat{\boldsymbol{\theta}}^{(m-1)}), g^{(m-1)})) \geqslant \gamma$. Then, we bound the number of queries $n^{(m)}$ in fidelity $m$. The training at fidelity $m$ stops when $\text{WINFREQ}(f(\cdot; \boldsymbol{\theta}_t), g^{(m)}) \geqslant \gamma$, that is, $\boldsymbol{\theta}_t = \hat{\boldsymbol{\theta}}^{(m)}$. We have

$$\|\hat{\boldsymbol{\theta}}^{(m)} - \hat{\boldsymbol{\theta}}^{(m-1)}\| \overset{(a)}{\leqslant} \alpha |\text{WINRATE}(f(\cdot; \hat{\boldsymbol{\theta}}^{(m)}), g^{(m)}) - \text{WINRATE}(f(\cdot; \hat{\boldsymbol{\theta}}^{(m-1)}), g^{(m)})|$$

$$\leqslant \alpha |\text{WINRATE}(f(\cdot; \hat{\boldsymbol{\theta}}^{(m)}), g^{(m)}) - \text{WINRATE}(f(\cdot; \hat{\boldsymbol{\theta}}^{(m-1)}), g^{(m-1)})|$$

$$+ \alpha |\text{WINRATE}(f(\cdot; \hat{\boldsymbol{\theta}}^{(m-1)}), g^{(m-1)}) - \text{WINRATE}(f(\cdot; \hat{\boldsymbol{\theta}}^{(m-1)}), g^{(m)})|$$

$$\text{with probability} \overset{(b)}{\leqslant} \alpha |\gamma + \text{rad} - \gamma + \text{rad}| + \alpha\beta \, \text{WINRATE}(g^{(m)}, g^{(m-1)})$$

$$= 2\alpha \cdot \text{rad} + \alpha\beta \, \text{WINRATE}(g^{(m)}, g^{(m-1)}),$$

where inequality (a) is due to Lemma 2, and inequality (b) is due to Lemma 1 and Assumption 3 holds with a probability of at least $1 - \delta \sum_{m \in \mathcal{M}} |\mathcal{D}^{(m)}|$.

With Assumption 4, we have

$$\mathcal{D}^{(m)} \leqslant B + n^{(m)} = O\left( \log \frac{\|\hat{\boldsymbol{\theta}}^{(m)} - \hat{\boldsymbol{\theta}}^{(m-1)}\|}{1 - \text{WINRATE}(f(\hat{\boldsymbol{\theta}}^{(m)}, g^{(m)}))} + B \right)$$

$$= O\left( \log \frac{2\alpha \cdot \text{rad} + \alpha\beta \, \text{WINRATE}(g^{(m)}, g^{(m-1)})}{1 - \gamma - \text{rad}} + B \right),$$

where the last equality is from the above derivation. $\qquad \square$

