# OpenReview forum: "Cost-Efficient Multi-Fidelity Alignment for LLMs"
_ICLR.cc/2025/Conference — Submitted to ICLR 2025_

### Official Review · Reviewer_FMv1 · 2024-10-24

**Soundness:** 2
**Presentation:** 2
**Contribution:** 2
**Rating:** 3
**Confidence:** 4

**Summary:**

The paper addresses the challenge of aligning LLMs with human preferences, a process that typically requires expensive human annotations. The authors propose a multi-fidelity alignment approach that leverages varying levels of response quality and associated costs to train the model more cost-effectively while preserving its performance. They provide both theoretical insights and empirical evidence supporting the efficacy of this method, showing that lower-quality alignments can still yield competitive performance compared to traditional alignment methods.

**Strengths:**

(1) The introduction of multi-fidelity alignment is a novel contribution that seeks to reduce the costs associated with human annotation while maintaining model performance.
(2) The paper offers both theoretical insights and empirical results that validate the effectiveness of the proposed method, strengthening the argument for its adoption.

**Weaknesses:**

(1) Lack of innovation: the proposed framework in Algorithm 2 seems very similar to the SPIN (ICML24), but the authors just replaced the ground true pair with the response generated by the expert and added the compare function.
(2) The compare function is not clear in the paper. Is it a prompt-based method? if so, please explain it more clearly.
(3) The metric is not convincing. Why authors do not experiment with the Alpacaevel2.0 benchmark or Arena-Hard benchmark?
(4) The writing is poor. the intro part of the win rate metic is redundant, authors should pay more attention to introducing their own contribution.
(5) Hyper-parameter setting seems tricky. There still a huge page remaining, so why not report the grid search result if you're already done?
(6) Why authors do not conduct experiments with other preference optimization methods like ORPO they have cited in the previous text?
(7) Some incorrect latex format in line 588 --  " Lemma ??"

**Questions:**

See the weakness section.

---

### Official Review · Reviewer_7CjS · 2024-10-27

**Soundness:** 2
**Presentation:** 2
**Contribution:** 3
**Rating:** 5
**Confidence:** 4

**Summary:**

The paper introduces an approach to reduce the alignment costs associated with post-training of LLMs. Traditional alignment methods require extensive human annotations to align the model's output with human preferences, which is both time-consuming and expensive. To address this, the authors propose a multi-fidelity alignment method that leverages responses of varying qualities and costs. The core idea is to use multiple levels of alignment, each with different quality and response-generating costs. For instance, high-fidelity alignment might involve human experts, while low-fidelity alignment could use less capable language models.

**Strengths:**

1) The paper is commendable for its thorough theoretical foundation, providing a rigorous mathematical formulation of the WINRATE metric and its application in preference-based training.

2) The paper presents a pioneering approach to significantly reduce the financial and resource costs associated with aligning LLMs to human preferences. By introducing multi-fidelity alignment, which incorporates responses of varying qualities and costs, the authors offer a practical solution to the prohibitive expenses typically required for human annotations.

**Weaknesses:**

My major concerns are as follows:

The paper generates DPO datasets using an iterative scheme across various alignment models, as demonstrated in Algorithm 2, and subsequently uses this dataset to train an LLM to achieve performance gains. However, how can the authors ensure that these performance improvements stem from the proposed theory? This process appears more akin to knowledge distillation or data synthesis. Furthermore, while the paper claims to achieve a “weak-to-strong” result, it inadequately elucidates the implications of this transformation within the alignment process.

In conclusion, the paper's aims are unclear. If the goal is to enhance “weak-to-strong” generalization, the authors should provide further exploration of this aspect. Conversely, if the aim is to improve alignment through multi-fidelity LLMs, a more detailed analysis comparing this approach to related works, such as knowledge distillation and data synthesis, would be beneficial.

Other comments:

1) This approach employs a strategy of guiding model training using a hierarchy of weak to strong models. However, one of the research goals this paper is to minimize the overhead associated with obtaining a response. Introducing decoding operations for external models may, in fact, increase this overhead, yet there are no experiments provided that demonstrate a reduction in overhead.

2) The authors opted to use three distinct series of models in their experiments. However, to more clearly distinguish performance differences, it would be more effective to utilize various sizes or versions of the same model series.

3) The algorithmic process presented in this paper involves successively selecting models from weak to strong to generate responses, comparing these with the responses of the training models to derive preference data, and subsequently conducting DPO training. The efficacy of this staged training process, which selects models in a weak-to-strong manner, requires validation through ablation experiments.

4) The paper currently lacks sufficient baselines. The authors should incorporate additional baseline experiments, such as DPO with publicly available or human-annotated datasets, to validate the effectiveness of the proposed method. Furthermore, testing the final model on widely recognized benchmarks, such as AlpacaEval, would improve the robustness of the evaluation.

**Questions:**

Other suggestions and typos:
1) Type error: “meetshuman’s” in Instruction Section.
2) Line 588: “Lemma ??”

Please refer to weaknesses for other questions.

---

### Official Review · Reviewer_gkDL · 2024-10-30

**Soundness:** 1
**Presentation:** 1
**Contribution:** 2
**Rating:** 3
**Confidence:** 3

**Summary:**

The paper proposes a new approach to reduce alignment costs considering multiple levels of alignment with different qualities and costs. The authors call this "multi-fidelity" alignment. A high fidelity alignment is of better quality or accuracy (and cost) than a low fidelity alignment. The paper present a mathematical formulation of the WinRate metric used in preference training, and a theoretical analysis for the cost complexity upper bound of the multi-fidelity alignment algorithm.

**Strengths:**

The idea of having a method that can reduce costs for performing preference optimization (or alignment) is very useful. Leveraging multiple types of preference signals according to their level of quality is also an interesting idea to explore the cost reduction.

**Weaknesses:**

The paper is not clear in some passages. It is not easy to understand what is the difference between step and iteration. The text should be rephrased to make this distinction more clear in Section 5. Furthermore, it is difficult to appreciate the results without some context on the choice of the dataset used for the experiments. There is no meaningful comparison with previous work which also makes it difficult to put the results in context.

**Questions:**

* What is the reason for choosing Vicuna and not another instruction tuned model?
* What is the motivation for choosing rm-static dataset and task rather than some other task for alignment?

---

### Meta-Review · Area_Chair_Go1R · 2024-12-17

**Metareview:**

The paper presents a novel approach to reduce alignment costs in the post-training of large language models (LLMs) through "multi-fidelity" alignment. This approach involves utilizing multiple levels of alignment with varying qualities and costs, where high-fidelity alignment offers better quality at a higher cost, and low-fidelity alignment involves less expensive methods, such as using less capable language models. By leveraging responses of different quality levels, the proposed approach aims to train models cost-effectively while maintaining performance. The paper includes both theoretical insights and empirical evidence, demonstrating that lower-quality alignments can still achieve competitive results compared to traditional alignment methods.

However, the reviewers find that the paper lacks novelty, and the authors need to clarify the motivation of the paper more explicitly.

In addition, the author did not reply.

**Additional Comments On Reviewer Discussion:**

there was no rebuttal

---

### Decision · Program_Chairs · 2025-01-22

Reject